Journal of Machine Learning Research 1 (2021) 1-48                    Submitted 7/21; Published 10/21

# Detecting genetic alterations in *BRAF* and *NTRK* as oncogenic drivers in digital pathology images: towards model generalization within and across multiple thyroid cohorts.

**Author1**                                                    EMAIL@EMAIL.COM
**Author2**                                                    EMAIL@EMAIL.COM
**Author3**                                                    EMAIL@EMAIL.COM
**Author4**                                                    EMAIL@EMAIL.COM
**Author5**                                                    EMAIL@EMAIL.COM
**Author6**                                                    EMAIL@EMAIL.COM
*Affiliation*

**Editor:** to be added

## Abstract

In this paper, we describe the machine learning problem of identifying different types of tumors based on digital pathology images. Given a set of Hematoxylin and Eosin (H&E) stained images of thyroid tumors, we train deep learning models to detect two known molecular oncogenic drivers: *BRAF* mutations and *NTRK* gene fusions. We implement an attention-based multiple instance learning (MIL) classifier and we assess its generalization within and across three independent cohorts. We find that the model can detect both oncogenic drivers with the MIL approach, however the problem remains challenging: our exhaustive evaluation scenarios exemplify unknown data drifts and batch effects in digital pathology as the model performance decreases when processing images from an unseen cohort. These findings highlight the necessity of rich and diverse datasets for training and evaluation as well as methods for domain-agnostic learning.

**Keywords:**    digital pathology, deep learning, domain transfer, *BRAF*, *NTRK*, oncogenic driver detection, WSI classification, attention multiple instance learning, histology, histopathology, H&E slide

## 1. Introduction

The timely identification of targetable gene mutations remains a roadblock for precision oncology. In view of the recent revolutionizing advances in oncogene-targeted therapies, advanced companion diagnostic technologies are of key importance. Current optimal clinical practice relies at large on immunohistochemistry (IHC) or Next Generation Sequencing (NGS) to identify gene alterations which determines the eligibility of a patient for such targeted treatments, cf. (Malone et al., 2020). Since the access to IHC and NGS strongly depends on the clinical center, many patients are deprived from the opportunity to benefit from the newly available targeted cancer treatments. Hence, the search for low-cost and highly available alternative methods has attracted the interest of different research communities. A promising direction of research are computational methods that reliably detect oncogenic drivers based on H&E image morphology. These approaches build upon the successes of machine learning and computer vision methods which pushed the bound-

aries for many problems in computational pathology, such as tumor detection, classification or grading, cell detection and counting, image segmentation or the extraction of human interpretable features, cf. (Laak et al., 2021).

In the study put forward by Fu et al. (2020), transfer learning with deep neural networks pretrained on classic computer vision tasks was employed to quantify histopathological patterns in whole slide images (WSI). Analyzing H&E images from within the TCGA database, it was shown that image-base features not only allow to classify different tissue types, but also correlate with various genetic alterations. In particular, mutations in the *BRAF* gene could be predicted thyroid tumors with high patch-level accuracy. In Anand et al. (2020) a multi-stage image classification pipeline is proposed for detecting an over-expression of the human epidermal growth factor receptor 2 (HER2) mutation in breast cancer based on H&E-stained images. The performance of the end-to-end pipeline was evaluated on a testset containing 26 cases from the same data source which resulted in an AUC of 0.82. In order to assess the generalizability of their approach on independent multi-site data, a testset comprising 45 cases from the TCGA-BRCA cohort was curated. The performance of the algorithm on this external data degraded and a patient-level AUC of 0.76 was obtained.

The phenomenon of computational pathology algorithms performing well within a data domain, i.e. test and training data originate from the same source, while exhibiting a significant degradation in performance when tested on data from another source is well-known, cf. (Campanella et al., 2019). Laak et al. (2021) point out that the validation strategies and data sets used in recent research papers are not representative of the type of data that is encountered in clinical practice. As key steps towards a better assessment of the diagnostic value of novel data-driven methods, they consider the aggregation of data sets that reflect data heterogeneity. Moreover, extended validation standards are proposed that cover data from multiple sources with a prospective study for final validation.

The study at hand serves different purposes. We present comprehensive experimental results concerning the detectability of *BRAF* mutations on thyroid tumors based on H&E-stained WSI by the means of multiple instance learning using weak labels (i.e. no annotations). We show that the proposed method can also be applied for the detection of *NTRK* fusions as oncogenic driver of thyroid tumors, although *NTRK* gene fusions are very rare. To that end a total of 802 H&E-stained WSI from three different sources were aggregated comprising 421 WSIs from tumors with *BRAF* mutations (i.e. V600E and other known mutations) and 23 WSIs from tumors with *NTRK* fusions (e.g. ETV6–NTRK3 or IRF2BP2–NTRK1). This data covers various sources of variation such as different sites from different countries, multiple scanners and staining protocols. The validation of our algorithms focuses on the effect of multi-domain data and in particular the generalizability of the models across different cohorts and data sources. While *BRAF* mutation is the most frequent oncogenic driver in thyroid cancer in around half of all cases (Agrawal et al., 2014; Brose et al., 2014), *NTRK* gene fusion is a rare genetic abnormality with frequency for thyroid cancer ranging from 2.3% to 3.4% in predominantly adult cohorts, cf. (Solomon et al., 2020; Lee et al., 2020; Pekova et al., 2021). Consequently, the development of a machine learning classifier that predicts the *NTRK* status is considerably more challenging since the model can only be trained on a very small number of samples with NTRK fusions. In order to quantify the differences in independent cohorts, we study several evaluation scenarios and

Table 1: Dataset overview. Data from each cohort was acquired and processed independently.

| Cohort | Access | Site Distribution | #WSI total | #WSI NTRK$^+$ | #WSI BRAF$^+$ | #Patches per WSI $\mu(\pm\sigma)$; min, max |
|--------|--------|-------------------|-----------|-----------|-----------|------------------------------|
| TCGA | public data | 11 sites across USA | 482 | 12 | 294 | 9538($\pm$4737); 355, 29215 |
| DEC | private data from internal study | material from multiple international sites | 224 | 4 | 58 | 9939($\pm$4372); 286, 22400 |
| ACQ | private data, acquired from biobank | material from several sites across USA | 100 | 7 | 69 | 2873($\pm$901); 1139, 5607 |

assess the model performance in various configurations using different cohorts for training, testing and holdout.

## 2. Cohorts

This study incorporates histology data from three different cohorts: TCGA, DEC and ACQ. Each cohort is treated as independent dataset which comprises H&E-stained whole slide images with tumor tissue from patients with thyroid cancer indication. The experimental protocol for H&E staining was different for each cohort and the images were processed in different labs and with different scanners. All images were scanned with $40\times$ magnification and each image went through a manual quality check by a qualified pathologist. Slides were excluded if either the pathologist rated the image quality to be below the clinical standards, or if the molecular status with respect to *BRAF* and *NTRK* could not be clarified. The rights of the data and sample donors of the cohorts were respected by e.g. checking the informed consent used. Further details of the three cohorts are described in Appendix A and in Table 1.

### 2.1 Data Split

Each cohort was split into training set ($\sim 50\%$), validation set ($\sim 20\%$) and test set ($\sim 30\%$) as described in Table 2. Within a pseudorandom process it was ensured that the splits are well balanced with respect to label distribution, sites, image size and gender. For ACQ, it was ensured that the images from the same patient were allocated to the same split. The other cohorts contained exactly one image per patient. The same data splits were used for the experiments on *NTRK* and *BRAF*.

## 3. Method

### 3.1 Preprocessing

WSI images were processed into non-overlapping patches of size $224 \times 224$ pixels covering an area of $156.8 \times 156.8\mu$m, see Figure 1. Patches containing less than 20% tissue material were excluded by a rule-based filtering. Table 1 shows the number of patches for each cohort

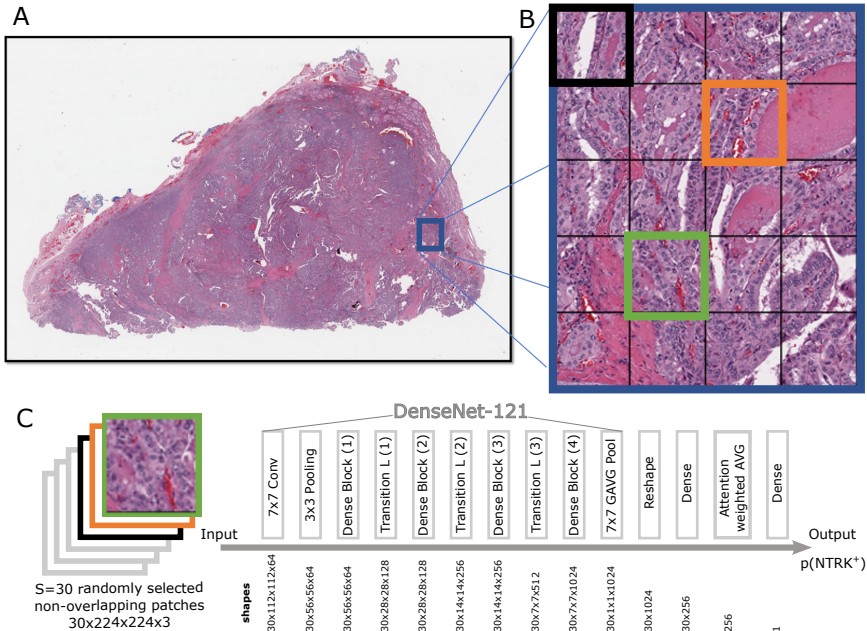

Figure 1: Thumbnail (A) of a whole slide image with thyroid cancer and the patch resolution (B). The Attention Multiple Instance Learning model with its shapes of the input, intermediate feature representations and output is depicted in (C).

after applying the data preprocessing. Color correction and image normalization was not performed.

## 3.2 Model

Multiple instance learning (MIL) algorithms are commonly applied for whole slide image classification problems with the underlying assumption that at least one small region of the slide contains morphological structure that correlates with the class label (Yao et al., 2020; Campanella et al., 2018; Ilse et al., 2020; Hashimoto et al., 2020). In analogy to Ilse et al. (2020) and Lu et al. (2021) we implemented a simple, yet effective attention-based deep multiple instance learning classifier (Ilse et al., 2018) with a DenseNet121 encoder (Huang et al., 2017). Hence, a classifier decision was provided for a bag of $S = 30$ randomly selected patches from one WSI. The architecture of the model and the shape of its intermediate feature representations are described in Figure 1C. The model has 7.3M trainable parameters, while 7.0M parameters of the DenseNet-121 encoder are initiated with ImageNet pretraining. Attention-weighted average pooling was applied on the patch embeddings in analogy to the implementation of Dippel et al. (2021): the model computes an attention weight $a_i$ for each patch $i \in \{1, ..., S\}$. Therefore, three consecutive convolution blocks (1D-convolution, batch norm, ReLU activation) are applied with decreasing number of filters $(256, 128, 64)$, followed by a final convolution with one filter and a sigmoid activation function. The entire attention module is trainable with the end-to-end model. To obtain a slide-level prediction

for one whole slide image $\mathcal{Q}$, the average classifier decision across $B = 90$ bags was computed. Thus, in total $B \times S = 2700$ randomly selected patches were processed to obtain a slide-level prediction for $\mathcal{Q}$.

### 3.3 Implementation details

Binary cross-entropy loss and an Adam optimizer with a learning rate of 0.001 were used to train the model. During the first 20 epochs the ImageNet-pretrained DenseNet-121 encoder was frozen and only the attention network as well as the classification head were trained. Afterwards all weights were unfrozen and jointly fine-tuned for 15 epochs with a reduced learning rate (factor 6). The model was evaluated after each epoch and the best-performing checkpoint was selected based on the AUC on the validation data. We used a batch size of 6, such that 180 ($6 \times 30$) image tiles were processed in one training step. One epoch consists of 300 training steps. Lightweight data augmentation was performed during model training: patches were randomly flipped along the x-axis and y-axis and a random zoom factor (range $\pm 5\%$) was applied to each patch. To account for class imbalances, importance sampling was performed on the training and validation data such that the rare class occurs in at least 25% of the samples. To reduce variance in the validation score and to improve the model selection process, the validation data was fixed during one experiment such that the model was evaluated on the same patches after each epoch.

## 4. Experiments

The problems of hidden batch effects and the lack of model generalization on external datasets have been widely discussed within the domain of digital pathology, cf. (Laak et al., 2021). Our experiments were designed to demonstrate and quantify the model performance for both, within-cohort and across-cohort evaluation scenarios. Table 2 describes the protocols (i.e. configuration of training, validation, test and holdout data) for six evaluation scenarios. For each of these scenarios we trained the binary classification models to detect the oncogenic driver. Thus, we trained models to detect the $BRAF$ status ($\text{BRAF}^+$ vs. $\text{BRAF}^-$) and we independently trained models to detect the $NTRK$ status ($\text{NTRK}^+$ vs. $\text{NTRK}^-$)[1]. Note that there are tumors that have no alterations in $BRAF$ and $NTRK$, thus $\text{BRAF}^-$ and $\text{NTRK}^-$. Patients with alternations in both genes were not present within the three cohorts. We explicitly did not combine the two problems into a multi-class classification problem to ensure a streamlined evaluation pipeline and an easy interpretation of the results, as $\text{BRAF}^+$ and $\text{NTRK}^+$ have substantially different prevalence in thyroid tumors.

The within-cohort generalization can be assessed through the model performance on the test set, while the across-cohort generalization can be assessed through the model performance on the holdout set. Note that testset results on scenario (1) and (2) were trained and evaluated solely on public TCGA data. This allows a direct comparison to literature and the results can be replicated without access to ACQ and DEC.

---

1. In the remainder of this article, $\text{BRAF}^+$ and $\text{NTRK}^+$ refers to a patient that has a mutation or fusion in the $BRAF$ and $NTRK$ gene respectively

When facing an extreme class imbalance and very low number samples, it can be advantageous to use the entire available data for training and to not perform model selection. This was done for scenarios referred to as "(train*)".

## 5. Results

Figure 3 reveals that *BRAF* can be detected as oncogenic driver consistently with an AUC of approx. 0.84 across all evaluation scenarios, if the model is evaluated on images from the same cohort that was used for training. Comparing the results obtained on the testset to the results on the holdout data, the AUC drops by approx. 0.1. Thus, the *BRAF* status can be determined with an AUC of $\geq 0.75$ for each holdout cohort. Note that all reported results are based on slide-level predictions, which is different to other studies that report patch-level or bag-level metrics (Campanella et al., 2019; Hashimoto et al., 2020).

The *NTRK* status can be assessed by the model with an AUC of more than 0.8 for each evaluation scenario if the model is evaluated on images from the same cohort that was used for training. However, the model performance drops significantly when evaluating on a holdout cohort. The model generalization even drops (close) to chance level for all scenarios except scenario (4) if the model is trained on the 50% train data split - see the orange bars. The results of the *NTRK*-detection problem reveal a clear pattern of "the more data used for training, the better the generalization on the external holdout data". The comparison between results for holdout(train*) and "holdout" reveals that the model performs significantly more accurate when model selection is skipped and all available images are used for training. Interestingly, this pattern was not observed for *BRAF* models, which indicates that a data driven model selection is beneficial if sufficient data is available from all classes.

Moreover, the TSNE visualizations in Figure 2 (right) illustrate the within-cohort similarities in the latent feature representation. Embeddings of the training and test data do not segregate from each other, which exemplifies that there is no substantial overfit to the training data, see Figure 2 (left).

The patient stratification plots in Figure 4 depict more details on the experimental results for scenario (4) for a real-world stratification use case. The differences in class distribution become more apparent with BRAF$^+$ being rather frequent and NTRK$^+$ very scarce.

## 6. Discussion

Oncogenic driver detection on H&E slides is possible, even if the model has only access to weak labels and no annotation. While Fu et al. (2020) described that *BRAF* can be detected in H&E images within the TCGA cohort, we confirm this finding using two external datasets as holdout cohorts. We show that *NTRK* gene fusions can be detected by a model, but the model performance is lower given the small number of NTRK$^+$ cases that are available in this study. However, our findings exemplify that also rare diseases can be detected and we see a great potential for further improvements. Given the three independent cohorts at hand, we were able to run exhaustive evaluation scenarios to study model generalization in a within-cohort and across-cohort setting. As expected, we find that the oncogenic driver detection is consistently more accurate on a testset which is from the same cohort that was used during

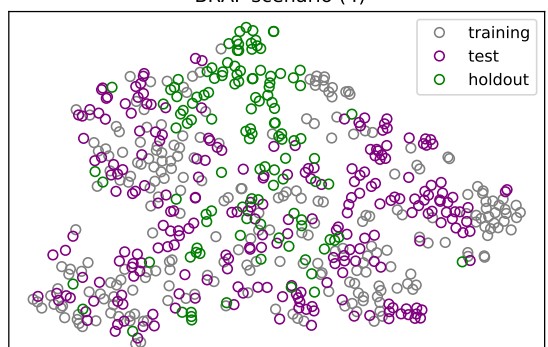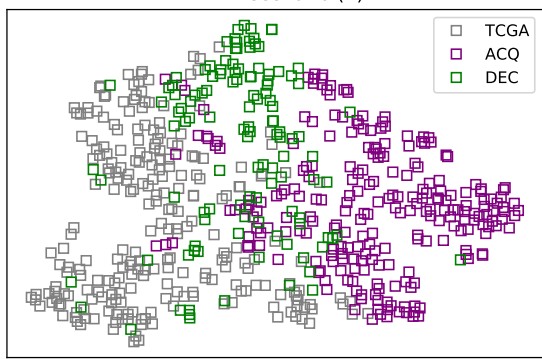

Figure 2: Domain shift in feature representations. Note that both plots show the same data with different marker labels.

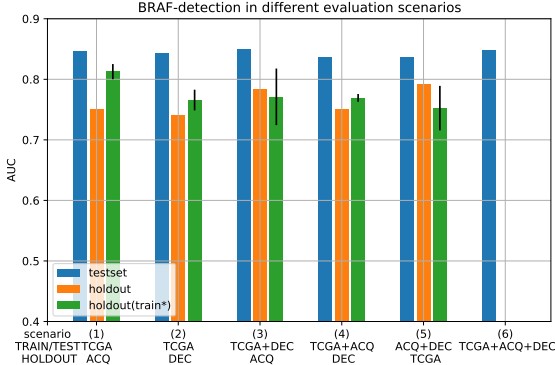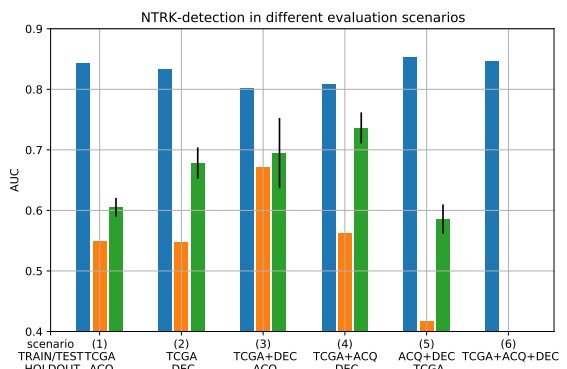

Figure 3: Domain-transfer results for different evaluation scenarios. Results on the testset assess within-cohort generalization, results on the holdout data quantify across-cohort model generalization. Further details for each scenario are provided in Table 2. The "holdout" and "holdout(train*)" results were generated on the same holdout dataset, but the "holdout(train*)" models were trained on the entire training data available. Model trainings for "holdout(train*)" were replicated 3 times with different seeds and the whiskers mark the observed standard deviation.

training. As it is widely discussed in literature (Laak et al., 2021; Lu et al., 2021), there are numerous hidden batch effects and site-specific characteristics that can be learnt by the model, which may not generalize to other cohorts. Interestingly, the (within-cohort) testset metric is not representative for the generalization capability on external holdout data: for the $BRAF$ detection problem, the testset AUC is consistently approx. 0.85 for all evaluation scenarios, while across-cohort holdout AUC ranges between 0.75 and 0.80. Also, for the $NTRK$ detection problem, the testset AUC is greater than 0.80 for each evaluation scenario and we observe a large variety for the AUC on the across-cohort holdout data ($0.55 - 0.73$).

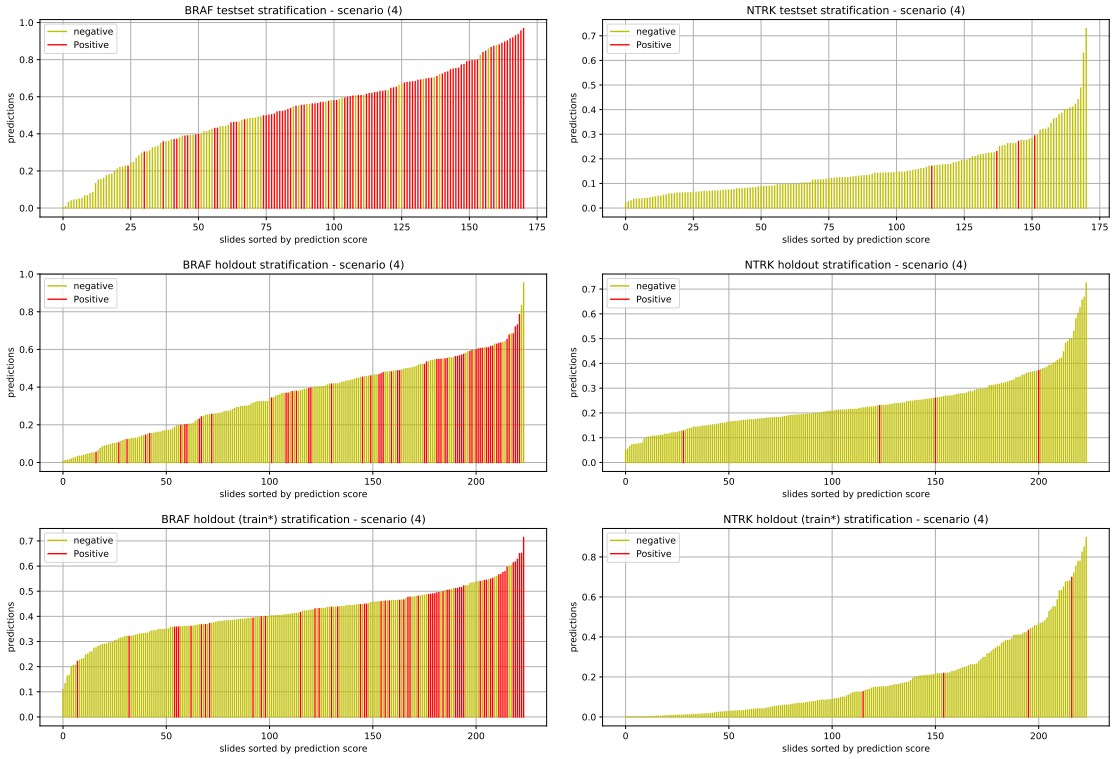

Figure 4: Patient stratification results for scenario (4). Plots show results for prediction of *BRAF* (left) and *NTRK* (right). For each stratification plot, the patients were ordered based on their model prediction and the color marks the molecular status. A perfect classifier would provide a plot with all yellow bars to the left and all red bars to the right.

## 7. Outlook

We plan to extend our analysis upon model generalization to additional data sources and cohorts. Furthermore, our algorithmic approach can be refined in various ways, e.g. by reflecting the mutual exclusivity of BRAF$^+$ and NRTK$^+$ in a common model, integrating methods to better cope with the *NTRK* low-label regime or incorporating prior knowledge obtained from semantic segmentations of the WSI. A deeper investigation of approaches covering multiple indications simultaneously could improve the overall model performance. Another future direction of research could aim at mitigating the effects of domain and concept shifts in data and labels across different sources. Next to methods for stain normalization (Janowczyk et al., 2017), a plurality of methods including techniques from metric learning, domain agnostic learning, transfer learning or continual learning have been proposed to tackle such problems in medical imaging, cf. (Lenga et al., 2020; Chen et al., 2019). We consider tailoring these methods to the task at hand or other computational pathology problems as fruitful starting point for further investigations.

## Appendix A. Appendix A: Details on datasets

- **TCGA** is the largest dataset which was generated by downloading all thyroid cancer from the publicly available *The Cancer Genome Atlas* (TCGA) database[2]. Samples in this cohort were acquired and processed at multiple sites. Slicing and HE staining was done centrally following one published SOP[3]. The molecular annotation, i.e. identification of *BRAF* and *NTRK* positives was done with NGS techniques. (Agrawal et al., 2014).

- **DEC** is an inhouse dataset from the 256-patient tumor genetic population of the global III trial[4] on the drug Sorafenib in thyroid cancer, cf. (Brose et al., 2014). *BRAF* status was determined using the Sequenom OncoCarta 1.0 panel, whereas *NTRK* gene fusions were detected via RNAseq (Capdevila et al., 2020). HE staining of the DEC samples was done centrally at a clinical research organization.

- **ACQ** is an in-house dataset which is based on tissue blocks and associated, limited clinical data obtained from a biotech biobank. The material was sourced from several sites throughout the USA and samples were processed centrally – slicing, staining, and scanning was done in an in-house lab facility. Molecular annotations for *BRAF* mutations and *NTRK* gene fusions were obtained using the FusionPlex Lung NGS panel (ArcherDx, Inc, Boulder, Colorado, USA). Note, that ACQ is the only dataset in which several images may originate from the same tumor: the 100 slides originate from 94 tumors, each tumor from a distinct patient.

Table 2: Details on the evaluation scenarios that were considered. Each row specifies the exact number of whole slides images which were used for one evaluation scenario and their distribution across the train, validation, (internal) test and (external) holdout dataset. Note that for scenario (6), all datasets were used for training, validation and testing such that there is no holdout set. Scenarios marked with train* used all available slides for training. Hence, model selection was not performed and the models were evaluated on the holdout dataset only.

| scenario | train set | | | | validation set | | | | test set | | | | holdout set | | |
|---|---|---|---|---|---|---|---|---|---|---|---|---|---|---|---|
| | $\sum$ | TCGA | DEC | ACQ | $\sum$ | TCGA | DEC | ACQ | $\sum$ | TCGA | DEC | ACQ | TCGA | DEC | ACQ |
| (1) | 239 | 239 | | | 94 | 94 | | | 149 | 149 | | | | | 100 |
| (2) | 239 | 239 | | | 94 | 94 | | | 149 | 149 | | | | 224 | |
| (3) | 341 | 239 | 102 | | 154 | 94 | 60 | | 211 | 149 | 62 | | | | 100 |
| (4) | 287 | 239 | | 48 | 124 | 94 | | 30 | 171 | 149 | | 22 | | 224 | |
| (5) | 150 | | 102 | 48 | 90 | | 60 | 30 | 84 | | 62 | 22 | 482 | | |
| (6) | 389 | 239 | 102 | 48 | 184 | 94 | 60 | 30 | 223 | 149 | 62 | 22 | | | |
| (1) train* | 482 | 482 | | | | | | | | | | | | | 100 |
| (2) train* | 482 | 482 | | | | | | | | | | | | 224 | |
| (3) train* | 706 | 482 | 224 | | | | | | | | | | | | 100 |
| (4) train* | 582 | 482 | 100 | | | | | | | | | | | 224 | |
| (5) train* | 324 | | 224 | 100 | | | | | | | | | 482 | | |

---

2. https://portal.gdc.cancer.gov/

3. https://brd.nci.nih.gov/brd/sop/download-pdf/1421

4. for details see https://clinicaltrials.gov/ct2/show/NCT00984282

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
