# OpenReview forum: "Detecting genetic alterations in BRAF and NTRK as oncogenic drivers in digital pathology images: towards model generalization within and across multiple thyroid cohorts."
_MICCAI.org/2021/Workshop/COMPAY — COMPAY 2021_

### Official Review · Reviewer_wAf3 · 2021-08-06
**The authors of this paper develop methods to detect BRAF mutation and NTRK gene fusions in thyroid cancer from H&E images using multiple instance learning.**

**Rating:** 6
**Confidence:** 4

**Review:**

The authors of this paper develop methods to detect BRAF mutation and NTRK gene fusions in thyroid cancer from H&E images using multiple instance learning. More specifically, the authors use three datasets to comprehensively run experiments to detect BRAF and NTRK. I like the quality of the paper - the paper was clear and easy to follow.
1. My main concern of this paper is the significance of the paper. I think the significance is to confirm the finding of BRAF detection in Fu et al. (2020) and to confirm NTRK detection is challenged to be generalized due to the lack of positive cases. The authors can explicitly state the significance of the paper and describe why it is different from other papers and meaningful to the community.
2. “(train*)” does not have a validation set to perform model selection. The authors need to describe how the model is selected without the validation set.
3. In Section 5, the author state that “the model generalization even drops (close) to chance level for all scenarios except scenario (4) if the model is trained on the 50% train data split – see the orange bars.” Did the authors mean scenario (3) whose orange bar reaches higher than 60%. If it was a typo, the end of Section 5 and the caption of Figure 4 need to be edited.
4. I have seen “cf” multiple times in the paper. What is “cf”?

---

### Official Review · Reviewer_vNiX · 2021-08-19
**A educational paper on H&E model generalization across independent cohorts using BRAF mutation and NTRK gene fusion as an examples**

**Rating:** 8
**Confidence:** 4

**Review:**

This educational paper provides an in depth discussion on model generalization in various experimental settings. The underlying network architectures and their specific implementation are well described, and the domain-transfer results nicely illustrated. While with BRAF the H&E analysis seems to provide robust predictions, the NTRK predictions seems to generalize far less. It would be interesting to see if a common multi-task network provides more robust predictions for NTRK, without sacrificing the performance for BRAF.

---

### Decision · Program_Chairs · 2021-08-25

Accept